# Alpha-Gal Syndrome—A Series of Cases with Different Clinical Pictures

**DOI:** 10.3390/ijms26178601

**Published:** 2025-09-04

**Authors:** Olga Branicka, Lesia Rozłucka, Radosław Gawlik, Joanna Glück

**Affiliations:** Department of Internal Disease, Allergology and Clinical Immunology, Medical University of Silesia in Katowice, 15 Poniatowskiego Street, 40-055 Katowice, Poland

**Keywords:** alpha-gal, meat allergy, delayed anaphylaxis

## Abstract

Alpha-gal syndrome (AGS) is an IgE-mediated allergy, triggered by a carbohydrate—galactose-α-1,3-galactose (α-Gal). AGS is marked by a delayed onset of symptoms, typically occurring 3–8 h after the ingestion of red meat or other mammalian-derived products. The primary risk factor is believed to be tick bites, which sensitize individuals through the introduction of α-Gal via tick saliva. Diagnosis of AGS is based on a combination of anamnesis and detection of α-Gal-specific IgE antibodies. We evaluated 28 patients with a history of unexplained anaphylaxis, angioedema, and/or urticaria, in whom the diagnostic work-up included the assessment of serum IgE specific to alpha-gal. Elevated alpha-gal-specific IgE levels were detected in five patients. Among them, four reported anaphylactic episodes following meat consumption. In three cases, symptoms developed during the evening or nighttime, typically 3 to 6 h after the last meal. One patient experienced anaphylaxis within one hour after a meal. Another patient presented angioedema up to 24 h after meat consumption and was also tested positive for specific IgE against beef and pork allergens. The AGS case series showed variability in clinical picture and time to reaction. In patients presenting idiopathic anaphylaxis and nonspecific symptoms after red meat consumption, AGS should be considered as a differential diagnosis.

## 1. Introduction

Alpha-gal syndrome (AGS) is an atypical form of IgE-mediated allergy, notable for being triggered not by a protein but by a carbohydrate—galactose-α-1,3-galactose (α-Gal)—present exclusively in non-primate mammalian tissues. It manifests as a frequently severe and characteristically delayed anaphylactic reaction to a carbohydrate allergen, distinguishing it from classical carbohydrate determinants (CCDs) [1,2].

In contrast to conventional food allergies, AGS is marked by a delayed onset of symptoms, typically occurring 3–8 h after the ingestion of red meat or other mammalian-derived products (e.g., beef, pork, lamb). This latency often contributes to diagnostic challenges and misinterpretation of symptoms [1,2,3,4,5]. Due to its mammalian origin, α-Gal-specific IgE antibodies may elicit allergic reactions not only to food products but also to pharmaceuticals of mammalian origin (e.g., antivenoms, gelatin-containing plasma expanders) and to biomedical products produced using mammalian cells or tissues [1,2].

The syndrome was first identified following episodes of anaphylaxis in patients administered the monoclonal antibody cetuximab. Subsequent investigations revealed that the cause was the presence of the α-Gal epitope in the Fab fragment of the drug [6]. The identification of specific IgE antibodies targeting α-Gal has since shaped the current understanding of AGS. The primary risk factor is believed to be tick bites, which sensitize individuals through the introduction of α-Gal via tick saliva, triggering a Th2-dependent IgE response [2].

Diagnosis of AGS is currently based on a combination of thorough clinical history—with particular attention to recent tick exposure—and detection of α-Gal-specific IgE antibodies, with symptom correlation [2]. In patients presenting with idiopathic anaphylaxis, AGS should be considered as a differential diagnosis.

In the general European population, robust and standardized data regarding the prevalence of clinically manifest AGS remain lacking. The most well-documented evidence pertains to rates of sensitization. Epidemiological data from Denmark indicate that the proportion of individuals with detectable specific IgE (sIgE) to galactose-α-1,3-galactose (≥0.1 kUA/L) increased from 1.3% in 1990 to 3.7% between 2012 and 2015, followed by a slight decline to 3.2% in 2016–2017. Importantly, these findings reflect sensitization rather than the prevalence of clinically symptomatic AGS [7]. In Germany, a study among foresters and hunters reported sensitization rates of 35.0% (≥0.10 kUA/L) and 19.3% (≥0.35 kUA/L), with approximately 8.6% of individuals with sIgE ≥ 0.35 kUA/L exhibiting clinical manifestations of red meat allergy or AGS [8]. These observations underscore that clinically apparent AGS is substantially less common than serological sensitization and is predominantly observed in populations with high exposure to tick bites.

The aim of the study was to analyze indications for detection of α-Gal-specific IgE antibodies and clinical pictures of patients with positive serum testing.

## 2. Case Reports

We analyzed the medical history of 28 patients who because of different reasons underwent detection of α-Gal-specific IgE antibodies during hospitalization in the Department of Allergology and Clinical Immunology of the Prof. K. Gibiński University Clinical Center in Katowice. Among the 22 (78.6%) patients hospitalized due to anaphylaxis, 10 (35.7%) were hospitalized due to idiopathic anaphylaxis, 5 (17.9%) due to food-induced anaphylaxis, 4 (14.3%) due to drug-induced anaphylaxis, 2 (7.1%) due to exercise-induced anaphylaxis, and 1 (3.57%) because of perioperative anaphylaxis. The remaining six (21.4%) patients were hospitalized due to symptoms of food allergy (5/28, 17.9%) or other symptoms (1/28, 4.57%) (Table 1, Figure 1).


Patient 1


A 21-year-old female patient with no comorbidities was admitted to the Department of Allergology and Clinical Immunology due to recurrent anaphylaxis reactions of unknown origin. Between May and August 2024, she had four episodes of anaphylactic reaction, with generalized urticaria, facial and tongue swelling, shortness of breath, abdominal pain with diarrhea and vomiting, and a drop in blood pressure, without loss of consciousness. The patient associated the reactions with meat consumption a few hours earlier and reported a history of a tick bite. Physical examination and basic laboratory tests showed no abnormalities. Serum IgE against alpha-gal was 2.96 kU/L; total IgE level was 151 kU/L; serum mast cell tryptase level was 13.6 μg/L. Allergen-specific IgE with food allergens was negative. The patient was diagnosed with alpha-gal syndrome.


Patient 2


A 46-year-old male patient with hypertension, hypercholesterolemia was admitted to the Department of Allergology and Clinical Immunology due to recurrent anaphylaxis reactions of unknown origin. On 16 November 2024, after consuming ground meat patties around 3:00 p.m., followed by sausage and alcohol around 6:00 p.m., he developed symptoms of anaphylaxis around midnight. The episode was characterized by generalized skin pruritus, shortness of breath, hypotension, and loss of consciousness. The patient was unaware of his history of a tick bite. Basic laboratory test results were within normal limits. Serum IgE against alpha-gal was 100 kU/L; total IgE level was 1056 kU/L; serum mast cell tryptase level was 3.07 μg/L. ALEX test results were as follows: Pen m 2 (prawn)—3.48; Api m 1 (honey bee)—1.87; Der p 10 (house dust mites)—5.13; Bla g 9 (cockroach)—2.22; and Ovi a (egg) 0.45 kL/U. The patient was diagnosed with alpha-gal syndrome.


Patient 3


A 58-year-old male patient with no comorbidities was admitted to the Department of Allergology and Clinical Immunology due to recurring anaphylaxis reactions of unknown origin. In 2018, after several hymenopterous insect stings, the patient presented the first episode of anaphylaxis. In the same year, at night, after awakening from sleep, the patient presented weakness, confusion, blood pressure decreased to 60/0 mmHg, and loss of consciousness. There were no skin lesions or bronchospasm during the episode. The patient was treated in the ED. Thereafter, two further episodes of anaphylaxis occurred with no identifiable causative agent. On admission to the Department, the patient’s general condition was good. The patient was unaware of his history of a tick bite. Physical examination revealed no abnormalities. Basic test results were within a normal range. Serum IgE against alpha-gal was 0.75 kU/L; serum mast cell tryptase level was 32 μg/L. ALEX revealed bee venom hypersensitivity with a positive Api m 10. The patient was diagnosed with mastocytosis, hypersensitivity to bee venom, and alpha-gal syndrome.


Patient 4


A 38-year-old male patient with no comorbidities was admitted to the Department of Allergology and Clinical Immunology due to recurring idiopathic anaphylaxis. Since 2022, the patient had had three anaphylactic reactions, presented as dizziness, weakness and massive urticaria. The symptoms disappeared spontaneously after about 30 min. The patient reported that, prior to the most recent anaphylaxis episode, he had eaten meat lasagna around 5 p.m. and around 9 p.m. went for a run. He did not link the previous episodes to any specific factor. The patient was unaware of his history of a tick bite. Physical examination and basic laboratory tests showed no abnormalities. Serum IgE against alpha-gal was 40 kU/L; total IgE level was 221 kU/L; serum mast cell tryptase level was 5.04 μg/L. Allergen-specific IgE with food allergens were negative. The patient was diagnosed with alpha-gal syndrome.


Patient 5


A 55-year-old male patient with no comorbidities was admitted to the Department of Allergology and Clinical Immunology due to recurrent massive urticaria and angioedema after beef and pork meat consumption a few hours earlier. The patient denied systemic symptoms. The patient was unaware of his history of a tick bite. Physical examination and basic laboratory tests showed no abnormalities. Serum IgE against alpha-gal was 100 kU/L; total IgE level was 509 kU/L; serum mast cell tryptase level was not determined. Positive allergen-specific IgE results were obtained for beef and pork, 7.61 kUA/L and 3.1 kUA/L, respectively, and negative for poultry meat. The patient was diagnosed with alpha-gal syndrome and red meat hypersensitivity.

## 3. Discussion

In the present case study, we evaluated 28 patients with a history of unexplained anaphylaxis, angioedema, and/or urticaria, in whom the diagnostic work-up included the assessment of serum IgE specific to alpha-gal (Figure 1). Elevated alpha-gal-specific IgE (sIgE) levels were detected in five patients, one woman and four men (Table 2, Figure 2). According to the evidence-based diagnostic and treatment algorithm proposed by Darsow et al., the diagnosis of alpha-gal syndrome (AGS) should involve a detailed clinical history—particularly concerning recent tick exposure—alongside testing for α-Gal-specific IgE and correlation of symptoms [1].

Allergic sensitization to AGS is initiated after repeated tick bites, during which salivary proteins and glycolipids carrying α-Gal epitopes are introduced into the host. Glycoprotein-bound α-Gal is processed by B cells and presented to T cells, while glycolipid-bound α-Gal may also activate iNKT cells, promoting an IL-4-rich environment that drives class-switch recombination and the production of α-Gal-specific IgE. These IgE antibodies bind to mast cells and basophils, priming them for subsequent reactions. Upon ingestion of mammalian meat, α-Gal associated with dietary lipids is incorporated into chylomicrons during intestinal lipid absorption. After a delay of several hours, chylomicrons bearing α-Gal enter the circulation and interact with IgE-coated effector cells. Cross-linking of IgE on mast cells and basophils induces degranulation and the release of mediators, leading to delayed systemic allergic reactions [9].

In our series of patients, specific IgE against alpha-gal ranged between a borderline value at 0.75 kU/L to very high values, exceeding 100 kU/L in two cases. Similarly, total IgE levels were also skewed from between 9.26 to as high as 1056. In all the patients, the peripheral blood eosinophil number was normal. Interestingly, in our study, the patient with a low alpha-gal IgE level (2.96 kU/L) presented recurrent idiopathic anaphylaxis; in the patient with diagnosed mastocytosis, the alpha-gal IgE level was positive, although as low as 0.75 kU/L; at the same time, in the patient with isolated angioedema, the alpha-gal IgE level was 100 kU/L [10]. Commins et al. demonstrated that patients with relatively low levels of specific IgE antibodies may experience severe reactions, whereas others with high titers remain subclinical or present with only mild symptoms. These findings imply that antibody affinity, epitope specificity, and effector mechanisms are likely more relevant than concentration alone [11].

Among the five patients with elevated sIgE, four reported anaphylactic episodes following the consumption of meat. In three cases, symptoms developed during the evening or nighttime, typically 3 to 6 h after the last meal. This temporal pattern aligns with the findings of Commins et al., who highlighted the delayed nature of AGS reactions as a major diagnostic obstacle. The latency between allergen ingestion and symptom onset often obscures the association with food intake, leading to underrecognition of the condition [4].

However, one patient (Patient 3) experienced anaphylaxis within one hour of eating. This atypically rapid reaction may be explained by the patient’s concurrent diagnosis of systemic mastocytosis. As Carter et al. have reported, AGS may be associated with clonal mast cell disorders, which can modify clinical presentation and increase reactivity to various triggers [12].

Another patient (Patient 5) presented with angioedema up to 24 h after meat consumption. This individual not only had elevated α-Gal-specific IgE but also tested positive for specific IgE against beef and pork allergens, suggesting a dual allergic mechanism. This mixed profile—consistent with both immediate-type hypersensitivity and AGS—is rare but noteworthy. As Commins et al. observed, the absence of immediate oral symptoms in AGS frequently results in skin prick tests showing wheals smaller than 4 mm, further complicating differentiation from classic IgE-mediated food allergies [4]. In turn, Fischer et al. demonstrated that meat-induced delayed anaphylaxis was found only in 8.6% of the investigated with alpha-gal IgE levels ≥ 0.35 kUA/L [8].

Mabelane et al. reported that alpha-gal IgE levels > 5.5 kU/L and alpha-gal IgE total IgE ratio > 2.12% are highly likely to result in clinically significant meat allergy (AUC 0.95). In our study, all the patients have a ratio above 2.12; only one patient has a ratio of 2 (patient 1), which is a borderline value. Differences in total IgE levels (Figure 2) may also play a role in the regulation of the humoral immune response. However, as previous studies suggest, they do not directly correlate with the severity of clinical symptoms in this particular syndrome [13].

Currently, the cornerstone of AGS management remains strict avoidance of α-Gal-containing substances—primarily red meat (beef, pork, lamb), organ meats, cetuximab, and gelatin-containing foods or pharmaceuticals. Acute allergic reactions should be treated with antihistamines and corticosteroids, with intramuscular epinephrine indicated in severe cases. MacDougall et al. have explored emerging therapeutic strategies, including anti-IgE monoclonal antibodies and immunotherapeutic approaches, though these remain largely investigational [1,14].

As AGS becomes increasingly recognized worldwide, improving clinical awareness and establishing standardized diagnostic protocols are crucial to reducing patient morbidity and alleviating the burden on healthcare systems.

At present, there is no causal treatment for AGS. The cornerstone of management is complete avoidance of red meat and other mammalian-derived products. Dairy products can usually be maintained in the diet unless symptoms persist despite meat avoidance. Approximately 15% of patients require elimination of dairy, and only ~5% must avoid gelatin [3]. Initial attempts at oral desensitization have been reported, which require daily consumption of portions of meat [15].

## 4. Conclusions

In patients presenting idiopathic anaphylaxis and nonspecific symptoms after red meat consumption, AGS should be considered as a differential diagnosis. There is a recommended algorithm for alpha-gal syndrome, published in 2024 [1]. It recommends obtaining a detailed history of past anaphylactic reactions and any potential tick bites, followed by laboratory testing, including specific IgE to alpha-gal, as well as skin-prick testing with meat. Symptoms of AGS may develop and be of a different kind regardless of the absolute value of specific IgE against alpha-gal.

## Figures and Tables

**Figure 1 ijms-26-08601-f001:**
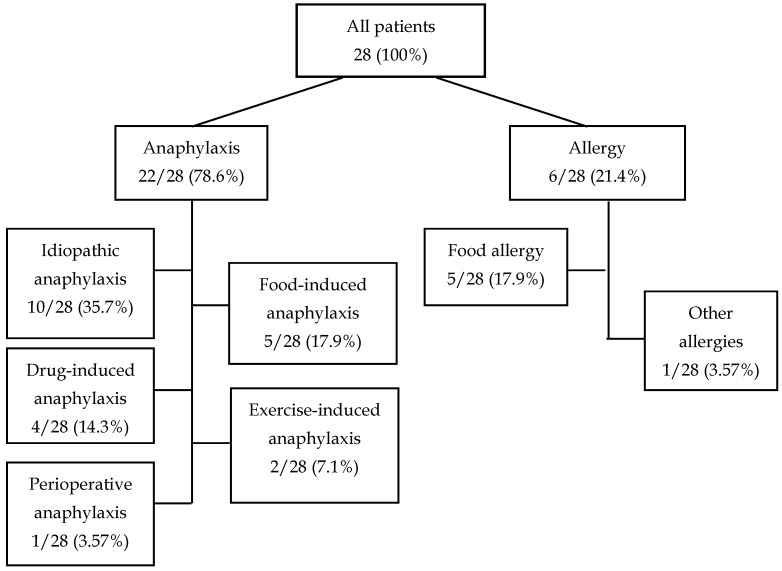
Classification of patients according to hospitalization reason.

**Figure 2 ijms-26-08601-f002:**
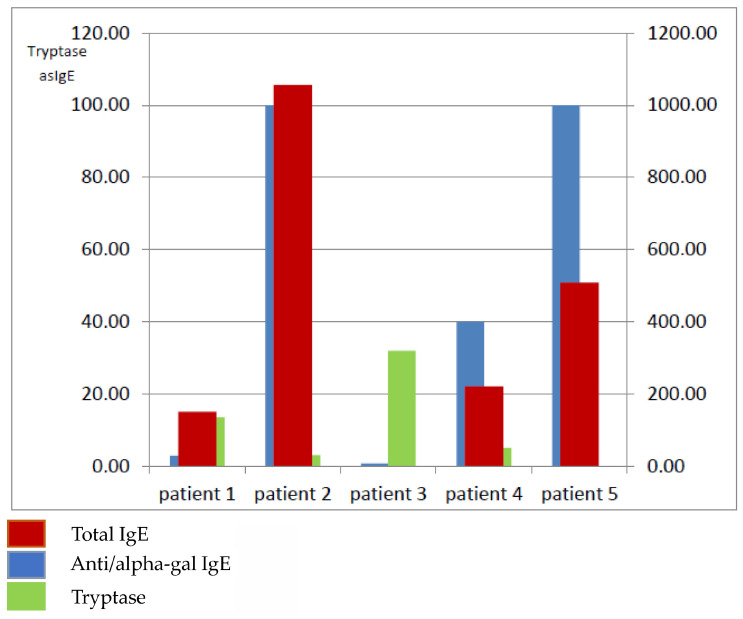
Values of tryptase, total IgE, and anti/alpha-gal IgE levels in all patients from the subgroup with positive anti-alpha-gal IgE.

**Table 1 ijms-26-08601-t001:** Characteristics of all patients and a subgroup with positive anti-alpha-gal IgE.

	All	Alpha-Gal (+)
n	28	5
Male, n (%)	13 (46)	4 (80)
Female, n (%)	15 (54)	1 (20)
Age (mean ± SD)	42.3 ± 12.6	44 ± 15
Atopic diseases, n (%)	14 (50)	1 (20)
Comorbid conditions, n (%)	12 (42)	1 (20)
Cardiovascular disease	5 (17)	1 (20)
Thyroid disease	6 (21)	0
One episode of anaphylaxis, n (%)	7 (25)	1 (20)
Episodes of anaphylaxis > 1, n (%)	14 (50)	4 (80)
Medical intervention, n (%)	16 (57)	5 (100)

**Table 2 ijms-26-08601-t002:** Characteristics of patients with alpha-gal syndrome.

	Gender	Age	BMI	Atopy	Symptoms	Number of Pisodes	Total IgE IU/mL [<87]	Tryptase µg/L [0–11.4]	Eosinophil Count, 10^3^/µL [0.04–0.4]	Alpha-Gal [>0.35]	Ratio as/IgE/tIgE
1.	F	21	21	Yes	Urticaria, angioedema, dyspnea, feeling of an obstruction in the throat, pressure drop, weakness	3	151	13.6	0.11	2.96	2.0
2.	M	46	31	No	Urticaria, angioedema, itching, dyspnea, pressure drop, loss of consciousness	1	1056	11.4	0.17	>100	9.5
3.	M	59	26	No	Dyspnea, weakness, dizziness, pressure drop, loss of consciousness	2	9.26	32	0.23	0.75	8.1
4.	M	39	24	No	Urticaria, weakness, loss of consciousness, dizziness	2	221	5.04	0.06	40	18.1
5.	M	55	37	No	Urticaria, angioedema, itching, erythema	3	509	-	-	>100	19.6

BMI—Body mass index; tIgE—total IgE; square brackets indicate normal ranges.

## Data Availability

The original contributions presented in this study are included in the article. Further inquiries can be directed to the corresponding author.

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
