# Peer review of "Alpha-Gal Syndrome—A Series of Cases with Different Clinical Pictures"

_ijms, 2025, doi:10.3390/ijms26178601_

Round 1

Reviewer 1 Report

Comments and Suggestions for Authors

The Case series by Branicka et al. submitted for review presents an interesting problem of Alpha-Gal Syndrome (AGS) in Polish population. The paper does a good job and addresses an interesting topic. It should be noted that this issue is quite new and there is no wide data in the available literature.

However, I feel that some aspect should be revised before the article could be published:

1. Table 1 – perhaps additional information such as age, comorbid conditions, diseases of atopic origin.

2 Table 2 – It is necessary to complete the data, i.e., reference ranges and units of the parameters being indicated; an explanation of the abbreviations used is provided below the table.

3. Case descriptions - is it possible to supplement the information with, for example, the time from food intake to the onset of symptoms, or whether all patients had previously been bitten by ticks.

4. Hence the work will also be read by non-allergologists, it might be worth explaining the shortcuts of allergens (e.g., Pen m, Api m etc.).

5. Discussion section – lines 201-204 – Is it recommended to perform tests with cetuximab or other pharmaceuticals in patients with a diagnosis of AGS before their use, or is their use unequivocally contraindicated?

6. Conclusions section – In this place, a diagram related to the authors' proposal for AGS diagnostics (e.g., what to pay attention to in the interview, what laboratory tests to order) would be useful for the readers.

7. You also do need to check the paper for typing errors, punctuation errors, spacing etc.

In my opinion, the paper is a solid contribution to the understanding of AGS, but by addressing these recommendations, the manuscript can be strengthened and the findings can be more robust and valuable to the scientific community.

Best regards.

Author Response

Response to Reviewer 1

Thank you very much for taking the time to review this manuscript. Please find the detailed responses below and the corresponding revisions, corrections highlighted in track changes in the re-submitted files.  

  1. Table 1 – perhaps additional information such as age, comorbid conditions, diseases of atopic origin.

Response: Corrected at the table.  

2 Table 2 – It is necessary to complete the data, i.e., reference ranges and units of the parameters being indicated; an explanation of the abbreviations used is provided below the table.

Response: Corrected at the table.

  1. Case descriptions - is it possible to supplement the information with, for example, the time from food intake to the onset of symptoms, or whether all patients had previously been bitten by ticks.

Response: Corrected at the manuscript.

  1. Hence the work will also be read by non-allergologists, it might be worth explaining the shortcuts of allergens (e.g., Pen m, Api m etc.).

Response: Corrected at the manuscript.

  1. Discussion section – lines 201-204 – Is it recommended to perform tests with cetuximab or other pharmaceuticals in patients with a diagnosis of AGS before their use, or is their use unequivocally contraindicated?

Response: In patients with AGS, the diagnostic evaluation for cetuximab hypersensitivity is not recommended.

  1. Conclusions section – In this place, a diagram related to the authors' proposal for AGS diagnostics (e.g., what to pay attention to in the interview, what laboratory tests to order) would be useful for the readers.

Response: Corrected at the manuscript.

  1. You also do need to check the paper for typing errors, punctuation errors, spacing etc.

Response: Corrected at the manuscript.

Reviewer 2 Report

Comments and Suggestions for Authors

see attached file

Author Response

Response to Reviewer 2

Thank you very much for taking the time to review this manuscript. Please find the detailed responses below and the corresponding revisions, corrections highlighted in track changes in the re-submitted files.  

  1. In discussion, the authors state that "AGS is becoming increasingly recognized worldwide" (Line 208). Therefore, it is recommended to provide data to support this statement in the Introduction. In particular, it is interesting to know the dynamics of AGS occurrence in European countries. It is necessary to indicate what treatment patients received, the outcome, and the prognosis. This will increase the practical value of the article.

Response: Corrected at the manuscript (introduction section).

  1. The International Journal of Molecular Sciences attracts readers who are interested in understanding pathological processes at the molecular level. Why not emphasize in the discussion the need for deciphering the molecular mechanism of AGS to obtain valuable information for patient care?

Response: Corrected at the manuscript (discussion).

  1. The conclusion is not entirely clear. On the basis of what diagnostic approach is it supposed to consider AGS as a differential diagnosis? Does the measurement of anti- alpha-gal IgE antibodies have diagnostic value if their concentration is not closely associated with clinical symptoms?

Response: Measurement of specific IgE (sIgE) alone is not sufficient for diagnosis; however, when interpreted in the context of a compatible clinical history—such as reproducible symptoms following consumption of red meat—it can provide supportive evidence for alpha-gal syndrome
